# Human Olfactory Bulb Neural Stem Cells (Hu-OBNSCs) Can Be Loaded with Paclitaxel and Used to Inhibit Glioblastoma Cell Growth

**DOI:** 10.3390/pharmaceutics11010045

**Published:** 2019-01-21

**Authors:** Hany E. Marei, Patrizia Casalbore, Asmaa Althani, Valentina Coccè, Carlo Cenciarelli, Giulio Alessandri, Anna T. Brini, Eugenio Parati, Gianpietro Bondiolotti, Augusto Pessina

**Affiliations:** 1Department of Cytology and Histology, Faculty of Veterinary Medicine, Mansoura University, Mansoura 35116, Egypt; 2Institute of Cell Biology and Neurobiology, National Research Council of Italy, 00015 Rome, Italy; patrizia.casalbore@cnr.it; 3Biomedical Research Center, Qatar University, Doha 2713, Qatar; aaja@qu.edu.qa; 4CRC StaMeTec, Department of Biomedical, Surgical and Dental Sciences, University of Milan, 20133 Milan, Italy; valentina.cocce@guest.unimi.it (V.C.); anna.brini@unimi.it (A.T.B.); augusto.pessina@unimi.it (A.P.); 5Institute of Translational Pharmacology, National Research Council of Italy, 00133 Rome, Italy; carlo.cenciarelli@ift.cnr.it; 6Cellular Neurobiology Laboratory, Department of Cerebrovascular Diseases, IRCCS Neurological Institute C. Besta, 20133 Milan, Italy; giulio.alessandri@istituto-besta.it (G.A.); eugenio.parati@istituto-besta.it (E.P.); 7IRCCS Istituto Ortopedico Galeazzi, 20161 Milan, Italy; 8Department of Medical Biotechnology and Translational Medicine, University of Milan, 20129 Milan, Italy; gianpietro.bondiolotti@unimi.it

**Keywords:** human olfactory bulb neural stem cells, paclitaxel, glioblastoma, stem cell-based therapy for glioblastoma, chemotherapy

## Abstract

Exploitation of the potential ability of human olfactory bulb (hOB) cells to carry, release, and deliver an effective, targeted anticancer therapy within the central nervous system (CNS) milieu remains elusive. Previous studies have demonstrated the marked ability of several types of stem cells (such as mesenchymal stem cells (MSCs) to carry and release different anti-cancer agents such as paclitaxel (PTX). Herein we investigate the ability of human olfactory bulb neural stem cells (Hu-OBNSCs) to carry and release paclitaxel, producing effective cytotoxic effects against cancer cells. We isolated Hu-OBNSCs from the hOB, uploaded them with PTX, and studied their potential cytotoxic effects against cancer cells in vitro. Interestingly, the Hu-OBNSCs displayed a five-fold increase in their resistance to the cytotoxicity of PTX, and the PTX-uploaded Hu-OBNSCs were able to inhibit proliferation and invasion, and to trigger marked cytotoxic effects on glioblastoma multiforme (GBM) cancer cells, and Human Caucasian fetal pancreatic adenocarcinoma 1 (CFPAC-1) in vitro. Despite their ability to resist the cytotoxic activity of PTX, the mechanism by which Hu-OBNSCs acquire resistance to PTX is not yet explained. Collectively our data indicate the ability of the Hu-OBNSCs to resist PTX, and to trigger effective cytotoxic effects against GBM cancer cells and CFPAC-1. This indicates their potential to be used as a carrier/vehicle for targeted anti-cancer therapy within the CNS.

## 1. Introduction

Targeting tumor cells and their associated microvessels by a drug that is toxic to them but of low toxicity to normal cells is the main goal of cancer chemotherapy. Several approaches have been used to selectively transport such drugs into the tumor environment.

Beside their potential application as an effective cellular therapy for central nervous system (CNS) degenerative and traumatic diseases, engineered neural stem cells (NSCs) might represent a new effective modality for cancer therapy [1].

The inherent ability of NSCs to hone into primary and metastatic tumor lesions within the nervous tissue might indicate their marked ability to serve as a promising delivery route for anti-cancer agents.

We have previously isolated NSCs from the human olfactory bulb (OB) and engrafted them into Alzheimer’s, Parkinson’s, and spinal cord injury rat models. The engrafted NSCs survived in the rat brain for more than eight weeks, and restored lost neuronal and glial cells [2,3,4].

The success of isolation of NSC from human OB brings forth great hope, not only for regenerative medicine, but also for brain tumor therapy.

Previous studies demonstrated a marked ability of mesenchymal stem cells to be uploaded with anticancer and anti-angiogenic drug such as Doxorubicin and Paclitaxel (PTX) without displaying signs of toxicity. Both mouse and human MSCs were able to release PTX and to be able to induce cytotoxic effects on tumor and endothelial cells in their proximity [5].

NSCs are more suitable for engraftment in the nervous tissues than MSCs, and thus we hypothesize that using NSC as a vehicle to deliver PTX might induce a more effective and more comprehensive therapeutic approach by using their antitumor and anti-angiogenic effects via the release of PTX. They may also be more effective through possible renewal of damaged neural tissues associated with the tumor microenvironment. This simple in vitro procedure does not include any genetic manipulation might be used for drug delivery against brain tumors (such as glioblastoma multiforme (GBM) the most malignant brain glioma). To the best of our knowledge, no specific study has yet been devoted to studying the therapeutic potential of human olfactory bulb neural stem cells (Hu-OBNSCs) loaded with PTX (Hu-OBNSCs-PTX) on glioblastoma multiforme (GBM). Therefore, the main objectives of the present study were: (1) to analyze the ability of Hu-OBNSCs-PTX to upload PTX; (2) to assess the ability of Hu-OBNSCs-PTX to home GBM cells, and induce anti-tumor activity. With these objectives in mind, green fluorescent protein (GFP) Hu-OBNSCs-PTX were loaded with PTX, and their potential anti-tumor activity was tested in vitro using the U87MG GBM cell line. We concluded based on this study that Hu-OBNSCs can successfully upload PTX without compromising their proliferation or differentiation activities; and that exposure of GBM cells to Hu-OBNSCs-PTX conditional media (CM) and lysate was associated with inhibition of tumor growth and pronounced antitumor activities in vitro. Therefore, Hu-OBNSCs-PTX may represent a new approach for chemotherapy of human GBM, targeting the proliferating cancer cells.

## 2. Materials and Methods

### 2.1. Ethical Statement

Informed consents were obtained from adults patients planned to undergo craniotomy at the Institute of Neurosurgery, Catholic University, Rome, Italy. IRB approval, to collect the OBs tissues, was obtained by the Catholic University. Hu-OBNSCs were isolated from different donors.

### 2.2. Human Olfactory Bulb NSCs Isolation, Culturing and Immunocytochemical Analysis

Human OBs were isolated following a previously described procedure [6]. Briefly, immediately after surgical removal, the OB tissues were minced and digested for 30 min at 37 °C in 0.1% papain. Cellular suspensions were cultured in DMEM/F12 (1:1) serum-free medium at 37 °C in a humidified atmosphere containing 5% carbon dioxide for several weeks till the appearance of cellular clusters, or neurospheres which were dissociated with Accutase.

### 2.3. Immunocytochemical Analysis

Undifferentiated neutrospheres were maintained in DMEM/F12 (1:1) serum-free medium containing basic fibroblast growth factor (bFGF) and epidermal growth factor (EGF). For differentiation, neutrospheres were dissociated into single cells and a monolayer culture was established by culturing the single cells on at a density of 2 × 10^4^ cells/cm^2^ onto pre-coated Lab-Tek chamber slides (NalgeNunc, Rochester, NY, USA), previously treated with Matrigel Matrix. The cells were differentiated into neurons, astrocytes and oligodendrocytes by culturing into a differentiation medium in which FGF and EGF were replaced with fetal bovine serum (FBS) at 1%. The cells were fixed in with 4% paraformaldehyde in PBS for 15 min at room temperature, permeabilized with Tris-HCl (0.1 M) + TritonX100 (0.25%) for 10 min, blocked with 10% normal goat serum and then incubated overnight at 4 °C with the primary antibody (anti-nestin, anti-glial fibrillary acidic protein, and anti-MAP2). Following washing, the cells were incubated with specific secondary antibodies for one hour at room temperature. The nuclei were stained with bisBenzimide H33258 diluted in PBS (0.2 µg/mL; SIGMA, St. Louis, MO, USA). Immunocytochemistry analysis was performed using fluorescent microscopy on an OLYMPUS Bx5 with Spot CCD Camera (Olympus Corporation BX 60 Fluorescence Microscope, Shinjuku, Tokyo, Japan).

### 2.4. Harvest of G-CSC Conditioned Medium (CM G-CSC)

Conditioned medium from glioblastoma cancer stem cells (CM-G-CSC) was collected by culturing the GBM neurospheres in DMEM/F12 (1:1) serum-free medium containing EGF, and human recombinant bFGF) as reported previously [7]. Next, the neutrospheres were dissociated with Accutase, and plated at the concentration of 1 × 10^6^ in adherent condition on Matrigel. After reaching 80% confluency, the culture medium was replaced with DMEM-F12 containing 1% BSA (10 mL). The G-CSC were incubated for 48 h, and the CM-G-CSC was collected and centrifuged at 600× *g* for 5 min, filtered through a 0.22 μm syringe filter, and conserved at 4 °C until use.

### 2.5. Cell Invasion Assay

For cell invasiveness, we have used a 24-well Transwell Permeable Support (8 μm pore size, Costar, Cambridge, MA, USA). The polycarbonate membranes of the upper compartment (insert) was coated with Matrigel (1.5 mg/mL). The human olfactory bulb cells and Warton’s Jelly mesenchymal stem cells (WJ-MSCs) (1 × 10^5^ cells/well) were seeded onto the Matrigel-coated cell culture permeable insert. The lower compartment of the Transwell system was filled with DMEM–F12 medium containing 1% and 5% BSA, and CM derived from glioblastoma cancer cells (CM G-CSC). The cells were incubated for 48 h at 37 °C in a 5% CO_2_ atmosphere to allow the cells to invade the matrix and migrate into the lower chamber. After the end of incubation, the cells migrated to the lower compartment were fixed in cold 96% ethanol for 15 min, washed three times with PBS and stained with 0.1% crystal violet in 2% ethanol for 20 min at room temperature. Using micro-plate reader the concentration of the solubilized crystal violet was assessed by determining the absorbance at 570 nm. Experiments were done in triplicates three times independently.

### 2.6. Sensitivity of Hu-OBNSCs1 and Hu-OBNSCs2 to Paclitaxel

Paclitaxel (PTX) for testing sensitivity and loading Hu-OBNSCs was kindly provided by Fresenius-Kabi, Verona, Italy. Cytotoxic effects of PTX on Hu-OBNSCs1 and Hu-OBNSCs2 were evaluated in 24-multiwell plates (Corning Incorporated, Corning, NY, USA) seeded at 25,000 cells/well in 0.5 mL/well of complete medium. After an incubation of 24 h in the presence of PTX (from 100 ng/mL to 10,000 ng/mL), the cells’ viability were evaluated by a colorimetric method (CellTiter 96^®^ AQ_ueous_ One Solution Cell Proliferation Assay (MTS), Promega.com). Absorbance at 490 nm was recorded using a plate reader.

### 2.7. Tumor Cells and Wharton’s Jelly Mesenchymal Stem Cells

The human glioblastoma cell line (U87MG) [8,9] and the human pancreatic adenocarcinoma cells (CFPAC-1) [10] were kindly provided by Centro Substrati Cellulari, ISZLER (Brescia, Italy). Cells were maintained by 1:5 weekly passages in Dulbecco’s Modified Eagle’s Medium (DMEM) High glucose and 10% Foetal bovine serum (FBS) (U87 MG), and Iscove modified Dulbecco’s medium (IMDM) and 10% FBS (CFPAC-1). All reagents were provided by Euroclone (Pero, Italy).

Human WJ-MSCs were isolated, characterized and cultured in Dulbecco’s Modified Eagle’s Medium Low Glucose in the presence of 10% FBS as reported [11]. All subsequent experiments were performed using these cells taken from passage 4.

### 2.8. Paclitaxel Loading of Human Olfactory Bulb Cells

Drug loading was performed according to a modification of a standardized operating procedure previously set up for MSCs derived from several tissues (bone marrow, adipose tissue and gingiva) [12,13,14,15]. Briefly, 5 × 10^5^ Hu-OBNSCs were exposed to 2 μg/mL PTX for 24 h. Then, the neurosphere cells were washed twice in Hank’s solution (HBSS, Euroclone, Pero, Italy). Paclitaxel-primed cells (hu-OBs/PTX) were then seeded in a 25 cm^2^ flask to release the drug. After 24 h, conditioned media (CM), (h-OBs/PTX CM) was collected.

To measure the amount of drug internalized, each set of paclitaxel-primed cells was washed two times with Hank’s solution (HBSS, Euroclone, Pero, Italy) and suspended in complete medium (1 × 10^6^ cells/mL). The cells were lysed by sonication (three cycles of 0.4 s pulse at 30% amplitude each) (Labsonic U Braun, Reichertshausen, Germany) and centrifuged at 2500× *g* for 10 min and the lysate collected. The conditioned media and lysates from both PTX primed cells (h-OBs/PTX/CM; h-OBs/PTX/LYS) were tested for their anti-proliferative activity on standard cancer cell line CFPAC-1 and U87MG cells. Conditioned media and lysates from un-primed h-OBs (h-OBs/CM; h-OBs/LYS) were used as negative controls

### 2.9. In Vitro Anticancer Assay

The effect of the CM, Lysates and pure Paclitaxel against tumor cell proliferation was evaluated in 96 multiwell plates (Sarstedt, Germany) by using as targets U87GM and CFPAC-1. Briefly, 1:2 serial dilutions of pure drug or CM and Lysates were prepared in 100 µL of culture medium/well that received 1000 tumor cells/well. Pure PTX was tested at concentrations from 50 to 0.39 ng/mL. Cell growth was evaluated by MTT assay (3-(4,5-dimethyl-2-thiazolyl)-2,5-diphenyl-2-H-tetrazolium) at 7 days of culture (37 °C in air and 5% CO_2_) [12,16]. The inhibitory concentrations (IC_50_ and IC_90_) were calculated by Reed & Muench formula [17].

The anti-tumor activities of CM and lysate were compared to that of pure PTX and expressed as PTX equivalent concentration (PEC), applying the following algorithm: PEC (ng/mL) = IC50PTX (ng/mL) × 100/V50 μL-CM or V50 μL-lysate. IC50 PTX is the concentration (ng/mL) of pure PTX producing 50% inhibition of CFPAC-1 or U87 MG proliferation; V50 μL-CM and V50 μL-lysate are, respectively, the volumes (μL/well) of CM and lysate able to give 50% inhibition of CFPAC-1 or U87 MG. Evaluation of single cell PTX release (PR) in the conditioned medium by primed MSCs was measured as the ratio between the total amount of drug (PEC × volume of conditioned medium) and the number of seeded cells: PR (pg/cell) = PEC CM (ng/mL) × 10^3^ × CM volume (mL)/number of seeded cells. To measure the amount of PTX retained by a single hOBNSCs (pg/cell), we applied the following equation: PEC LYSATE (ng/mL) × 10^3^ × lysate volume (mL)/number of lysed cells.

### 2.10. High Performance Liquid Chromatogrphy (HPLC) Analysis

To evaluate the amount of paclitaxel in cell culture medium (400 µL) and in cells lysate (80 µL) was developed a reversed phase HPLC method. The samples were mixed (1/4 *v*/*v*) with ethyl acetate in a vortex for 5 min and centrifuged; the supernatant was dried under vacuum (Rotavapor R 110, Assago, Italy) and the residue reconstituted with 70 µL of mobile phase, filtered through 0.2 μm nylon filters (Phenomenex, Phenex-NY 4 mm). Then an aliquot of 40 μL was injected in HPLC. The chromatographic system was an Agilent 1100 Series machine (Agilent Technologies, Inc., Santa Clara, CA, USA) equipped with a nucleodur EC C_18_ column, 4.6 × 150 mm, 5 μm particle sizes (Macherey-Nagel, Pozzuoli, Italy). It was operated at 30 °C in isocratic mode in the presence of acetonitrile and 0.1 M Ammonium Acetate (50/50 *v*/*v*) as the mobile phase; that was pumped at rate of 0.7 mL/min, with eluent monitoring using a UV-Visible DAD detector (Agilent Techn, Santa Clara, CA, USA) at 238 nm. The paclitaxel retention time was 12.7 min. A calibration curve (10-25-50-100 ng) in drug free CM/lysate was prepared and processed with the samples and used to quantify paclitaxel (y = 0.5714x − 4; *r*^2^ = 0.9997). The extraction recovery of paclitaxel measured in calibration curve was 75%.

### 2.11. Statistical Analysis

Data are reported as mean ± standard deviation (SD). A Student’s t-test was performed to compare mean values by using GRAPHPADINSTAT program V.3, 1997 (GraphPad Software Inc., San Diego, CA, USA). Differences were considered statistically significant with a *p* value ≤ 0.05. Excel 2013 software (Microsoft, Inc., Redmond, WA, USA) was used for to study the linearity of response and the correlation according to the regression analysis.

## 3. Results

### 3.1. Morphology of Hu-OBNSCs Neurospheres, Immunecytochemical Analysis in Proliferation and Differentiation Conditions

Hu-OBNSCs were grown in suspension aggregates as neurospheres and cultured in proliferation medium supplemented with the mitogens EGF and basic fibroblast growth factor (bFGF) (Figure 1A). In the same culturing conditions these neurospheres showed high immunoreactivity for nestin, a specific intermediate filament protein, indicating their undifferentiated state (Figure 1B).

In differentiation conditions obtained with growth factor replacement from culture medium and adding 1% foetal bovine serum, morphological changes were evident. In fact, Hu-OBNSCs proliferation began to slow and cellular processes became longer, more branched, and tended to be interconnected with those of adjacent cells. Seven days later in these culture conditions astrocytes and neurons appeared. A considerable decrease in the number of nestin positive cells was observed (data not shown) and a concomitant increase of cells immunoreactive for GFAP (a specific marker of astrocytes), and for MAP-2 (Microtubule associated protein-2, a specific neuron marker) were identified. (Figure 1C). These properties gave the isolated Hu-OBNSCs criteria suggestive of neural stem cells (NSCs), and therefore these cells were considered to have met the morphological and immunocytological criteria of our previously isolated OBNSCs.

### 3.2. Invasion Ability of Hu-OBNSCs Versus WJ-MSCs

A transwell invasion assay was performed to determine the invasion ability of Hu-OBNSCs in comparison to WJ-MSCs. As is well known, the latter cellular populations have intrinsic migration properties towards tumor sites and are considered attractive vehicles for delivering of anticancer components.

The invasive properties of Hu-OBNSCs2 compared to WJ-MSC is shown in Figure 1D. We demonstrated a moderate increase of invasive potential by Hu-OBNSCs2 in response to CM G-CSCs. This was comparable to that observed in WJ-NSC versus the same CM G-CSCs. Moreover, we observed an evident inhibitory effect of the presence of serum (DMEM + FCS) on migration and invasion of Hu-OBNSCs2. This is due to the fact Hu-OBNSCs2 cells in this culturing condition were more differentiated. Conversely, WJ-MSCs showed very high invasive capacity in response to serum that represents proliferative induction.

The same the invasive properties of Hu-OBNSCs1 compared to WJ-MSC are shown in Figure 1E. We observed a very high invasion capacity of Hu-OBNSCs1 versus CM G-CSC compared to that observed in WJ-MSCs versus the same CM G-CSCs. As mentioned earlier for Hu-OBNSCs2 cells, WJ-MSCs displayed a very high ability to invade in the presence of serum, whereas Hu-OBNSCs1 cells markedly decreased their invasion ability in the same culture conditions.

### 3.3. Sensitivity of Hu-OBNSCs to Paclitaxel

The cytotoxic effect of PTX was evaluated on two different lines of Hu-OBNSCs (Hu-OBNSCs1 and Hu-OBNSCs2) (Figure 2). As shown by the histograms, increasing the concentration of PTX did not affect cell viability in either cell line until 4000 ng/mL concentration was reached. In Hu-OBNSCs2 a 50% reduction of cell viability at a concentration of 10,000 ng/mL was observed. In Hu-OBNSCs1 at a concentration of 10,000 ng/mL a 60% reduction of cell viability was noted. This confirmed that we could apply the standard PTX loading procedure at 2000 ng/mL set up for MSCs [12].

### 3.4. Efficiency of PTX Incorporation and Anticancer Activity of PTX Released by Primed h-OB Cells

The ability of Hu-OB to incorporate PTX was confirmed by priming the cells with 2000 ng/mL and evaluating the anticancer activity of the cell lysate (LYS Hu-OBNSCs/PTX). The drug releasing capacity was checked by testing the conditioned medium of primed cells (CM Hu-OBNSCs/PTX) after 24 h of their subculture. The anticancer activity was checked against the cancer cell line CFPAC-1 used as a target standard to validate the drug uptake and release by MSCs. The treatment of CFPAC-1 with pure PTX allowed us to define a standard dose–response curve of inhibition on which, according to a biological dosage assay, the amount of PTX in both cell lysate and CM was estimated [18]. Both LYS Hu-OBNSCs/PTX and CM Hu-OBNSCs/PTX inhibited CFPAC-1 cancer cell proliferation according to a dose–response kinetic that was normalized on the basal inhibition kinetics produced by lysates and conditioned medium of untreated Hu-OBNSCs. A significant linear regression slope (*p* < 0.001) and a coefficient of correlation (*r*^2^) of 0.61 and 0.98 were reported (Figure 3). The biological dosage determined by comparing the activity of LYS Hu-OBNSCs/PTX (V50) with the IC50 value of pure PTX (see box of Figure 3) indicated that PTX primed Hu-OBNSCs incorporated 0.19 ± 0.07 pg of drug per cell, and that after 24 h of sub culture about 52% of the drug incorporated was released from the cells (CM Hu-OBNSCs/PTX).

### 3.5. In Vitro Efficacy Against Human Glioblastoma Cells

Both the LYS Hu-OBNSCs/PTX and CM Hu-OBNSCs/PTX were tested on human glioblastoma cells U87GM (Figure 4A). An insignificant inhibition was produced by lysates and conditioned medium from untreated Hu-OBNSCs against U87GM, while LYS Hu-OBNSCs/PTX and CM Hu-OBNSCs/PTX produced a dramatic inhibition of cancer cell proliferation. Regression analysis showed a significant slope (*p* < 0.001) with *r*^2^ of 0.86 and 0.82 respectively (Figure 4B). In the histogram (Figure 4C) the inhibitory effect on tumor growth is reported as the number of PTX loaded Hu-OBNSCs, to better evidence their delivery capacity. Based on the amount of drug incorporated by Hu-OBNSCs we estimated that 50% inhibition of U87GM proliferation corresponded to 3228 cells/well.

### 3.6. HPLC Analysis of PTX Incorporated and Released by Hu-OBNSCs

The drug incorporation and the release of PTX by primed Hu-OBNSCs was confirmed by HPLC analysis (Figure 5). Standard HPLC chromatograms (1.000 ng/mL of PTX) showed that the drug was eluted with a peak at 12.7 min (Figure 5A). An identical peak retention time for PTX was eluted by processing both lysate and CM from Hu-OBNSCs loaded with PTX (Figure 5B,C). HPLC analysis revealed the presence of other nonspecific peaks (at 2.5–4 min) due to compounds produced by cells that however do not correlate with the presence of PTX; these peaks are also present in the chromatogram of control lysate and medium (Figure 5D,E). The presence of the main PTX metabolite (6 alpha-hydroxypaclitaxel) normally eluted at 5.5 min and of others PTX metabolites can be excluded [19].

## 4. Discussion

In the present study, we demonstrated that Hu-OBNSCs can be uploaded with and can release PTX in a concentration that was able effectively to inhibit the proliferation of human glioblastoma cells (U87GM) and CFPAC-1 in vitro. This crucial observation qualifies Hu-OB as a promising candidate for cell-based therapy against brain cancers such as GBM. Similar results were previously demonstrated for MSCs isolated from the amniotic membrane of the human term placenta: they were resistant to PTX, and inhibited tumor cell proliferation per se under specific culture conditions in vitro [20].

The morphological and immuocytochemical properties of Hu-OBNSCs cells that were identified in the present study were suggestive of NSC, in fact they formed nestin-positive aggregates/neurospheres in proliferation medium that had been supplemented with the mitogens EGF and bFGF. Removal of the mitogen triggered their differentiation into GFAP-positive astrocytes and MAP-2-positive neurons. These findings are in harmony with our previous studies [3,4,21,22,23,24,25,26], and clearly indicate the stem cell nature of isolated Hu-OBNSCs, based on their multipotent potential and their ability to differentiate into the different cells types forming nervous tissues.

Next, we tested the invasion ability of Hu-OBNSCs versus WJ-MSCs. WJ-MSCs were selected due to their previously demonstrated intrinsic migration properties towards tumor sites [13]. In comparison to WJ-MSCs, the Hu-OBNSCs showed a moderate increase in migration and invasion potential during their proliferation phase. However, in contrast to WJ-MSCs, the migration and invasion ability of Hu-OBNSCs toward G-CSCs were markedly inhibited following their differentiation into astrocytes and neurons in the presence of serum. Taken together, these findings demonstrate that Hu-OBNSCs exhibit migratory and invasive behaviour towards the human glioblastoma microenvironment that is as good as or better than that observed for WJ-MSC. This suggests that Hu-OBNSCs can be recruited as therapeutic agents for preclinical studies.

Next, we thought to determine the loading level at which Hu-OBNSCs are resistant to PTX, and whether or not Hu-OBNSCs-PTX are able to trigger anti-tumor effects against target cancer cells. The viability and proliferation ability of Hu-OBNSCs was not compromised until a 4000 ng/mL concentration of PTX, and the IC50 (at which 50% of Hu-OBNSCs displayed reduction of cell viability) was estimated to be 10,000 ng/mL. Based on the estimated IC50 of PTX, we conclude that the standard PTX loading procedure for Hu-OBNSCs is 2.000 ng/mL, the same as previously estimated for MSCs [12]. Previous studies have demonstrated that MSCs isolated from bone marrow [12], adipose tissue [18], and dermal fibroblasts [27] are resistant to PTX, with a range of IC50 values between 34.85–659.12 ng/mL. This indicates the marked resistance of Hu-OBNSCs to PTX compared to MSC isolated from different tissue sources. It has been reported that most PTX was incorporated into MSC within 48 h, and the cells were able to release PTX for to 120 h after priming [13].

The ability of Hu-OBNSCs to resist a PTX concentration of 10,000 ng/mL is remarkable, particularly when compared to the ability of cancer cells (such as GBM) to resist such high concentration levels. Therefore, we next tested the effects of Hu-OBNSCs-PTX on human glioblastoma cells U87GM. Hu-OBNSCs-PTX showed pronounced inhibition of U87GM proliferation and tumor growth. To provide convincing evidence about their delivery capacity, we estimated that the demonstrated 50% inhibition of U87GM proliferation corresponded to 3228 Hu-OBNSCs/well. Human MSCs have previously been shown to exert strong anti-tumor [28] and anti-angiogenic activities [29], and currently are used to treat advanced solid tumors [30,31]. Based on the results of the present study, we estimate that if translated in vivo, the injection of 10^6^ Hu-OBNSC/PTX cells into a tumor mass of 1 cm^3^ could result in a PTX concentration of about 190 ng/mL, an amount that corresponds to 24.3 times the IC50 value found for PTX against glioblastoma cells in vitro. This implies the clear potential of Hu-OBNSCs-PTX for cell-based therapy of GBM. To confirm the ability of Hu-OBNSCs to release PTX, we used HPLC analysis. PTX was eluted with a peak at 12.7 min in both lysate and CM from Hu-OBNSCs loaded with PTX. The presence of the main PTX metabolite (6 alpha-hydroxypaclitaxel), normally eluted at 5.5 min and of other PTX metabolites can be excluded [19].

The mechanism by which Hu-OBNSCs are able to resist a concentration of 10,000 ng/mL PTX without compromising their proliferation ability and viability is still a matter of debate. This becomes even more mysterious in view of the assumption that an effective level for destruction of GBM cells is about 2000 ng/mL PTX. This means that the ability of Hu-OBNSCs to resist PTX is five more times higher than cancer cells. The mechanism by which PTX induces its cytotoxic effect in cancer cells, based on information from previously studies implies that the major cytotoxic effect of PTX is its ability to bind to microtubules [28]. Whether or this is also the case for Hu-OBNSCs is not yet determined. To the best of our knowledge, ours is the first study to highlight the marked ability of Hu-OBNSCs to resist PTX. Previous studies have tried to decipher the mechanism of resistance of some stem cells such as MSCs to PTX. Such studies concentrated on the role of certain drug-efflux transporters such as P-gp on the ability of human BM-MSCs to resist PTX [28]. Such drug-efflux transporters have been previously found in placental cells and were thought to play a role in bioavailability of different drugs between the fetal and maternal compartments [32,33]. Other mechanisms by which normal or cancer cells can display resistance to drug/chemotherapeutic agents have been suggested. The drug resistance of some cancer cells (such as breast cancer) has been attributed to their ability to modulate/upregulate the expression of breast cancer resistant protein (BCRP) [34]. Moreover, the presence of the drug-efflux P-gp protein has been demonstrated in placental syncytiotrophoblast cells as well [35,36]. Whether or not the drug-efflux P-gp protein (or a similar protein) is implicated in the marked resistance of Hu-OBNSCs to PTX is a matter that needs further investigation, because previous studies have asserted that despite its expression in human placental cells, P-gp protein does not seem to have a role in PTX transport [37]. Other drug-efflux enzymes such as CYP1A and CYP2E1 [38] have been identified in placental cells, although until now the role of theses enzymes in the transport of PTX has not been clarified.

The five-fold increase in the ability of Hu-OBNSCs to resist PTX compared to MSCs qualifies Hu-OBNSCs as promising candidates to target cancer cells especially within the CNS milieu. These exciting findings necessitate the initiation of further studies with the aim of highlighting other potential mechanism(s) underpinning the high resistance ability of Hu-OBNSCs. These might include tubulin gene mutations [39]. The presence of different tubulin isotypes [40], factors affecting microtubule liability [41], the possible role of other proteins such as in surviving [42], and proteins involved in controlling cell cycle regulation and cell survival may also be considered.

Although the mechanism by which Hu-OBNSCs take up and release PTX is still in need of further investigation, our data complements previous information from the studies of Pessina et al. [13], who were the first to demonstrate that through the simple process of in vitro priming, MSC incorporate PTX in an amount sufficient to inhibit tumor cell proliferation in vitro. Herein we report for the first time the five-fold higher ability of Hu-OBNSCs versus MSCs to resist PTX, together with their ability to inhibit the proliferation and invasion ability of GBM CSC. A lot of studies are directed to improve the drug bioavailability within many systems (liposome, protein carrier, beads, etc.). Many of them did not have great success. The “cell-mediated delivery” could be in effect more difficult in terms of ethical approval issues. However, we think that this opens a new way in at least for three aspects: (1) NSC may be integrated better with the CNS tumor environment; (2) Exploitation of the NSC homing potential toward CNS cancer cells; and (3) the possible production of drug associated with EVs/exosomes isolated from NSC. Taken together, these findings show that Hu-OBNSCs exhibit migratory and invasive behavior towards the human glioblastoma microenvironment similar to or better than that of WJ-MSC. Moreover, the Hu-OBNSCs seems to offer a significant advantage over MSCs in term of their potential use to target different tumors with the CNS milieu. That would be expected to expedite preclinical and clinical translation of Hu-OBNSCs-PTX as a vehicle for targeted chemotherapy against different brain and CNS cancers. Of course, this topic needs to be focused with in vivo studies that our laboratories hope to perform in the near future.

## 5. Conclusions

The ability of Hu-OBNSCs to resist and release PTX and to trigger PTX-mediated cytotoxic effects on GBM cancer cells has been studied in vitro. Hu-OBNSCs displayed a five-fold increase in their resistance to the cytotoxic PTX. Moreover, they were able to inhibit the proliferation and invasion of GMB, and CFPAC-1 in vitro. Despite their marked ability to resist the cytotoxic activity of PTX, the mechanism by which Hu-OBNSCs acquire resistance to PTX has not yet been elucidated. Taken together, the data of the present study clearly qualifies Hu-OBNSCs to be tested as carrier vehicles for targeted anti-cancer therapy within the CNS. Of course, this proof of concept needs to be confirmed in vivo in order to demonstrate the efficacy on tumor progression for possible application in the setting of advanced cell therapies.

## Figures and Tables

**Figure 1 pharmaceutics-11-00045-f001:**
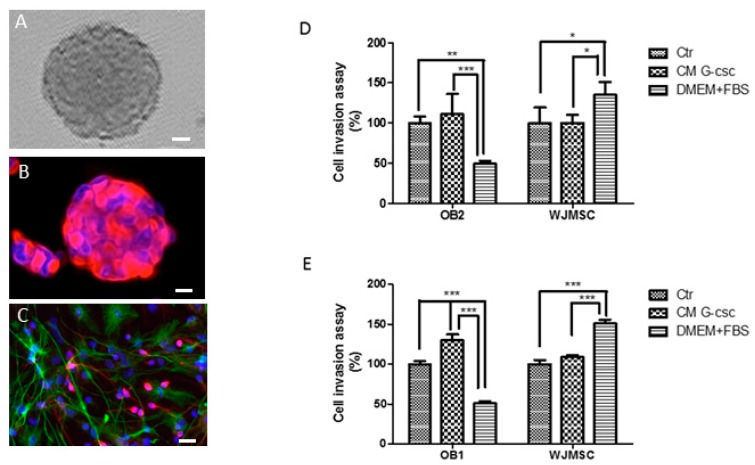
Morphology of human olfactory bulb (OB) neurospheres, immunocytochemical analysis and comparison of invasion ability of Hu-OBNSCs and WJ-MSC in different conditions. (**A**) Phase contrast picture shows a typical olfactory bulb (OB) neurosphere; the bar value is 25 µm. (**B**) In proliferative culturing condition OBs neurospheres were stained with marker anti-nestin (red). Cellular nuclei were counterstained with Hoechst 33258 (blue); the bar value is 25 µm. (**C**) At immunocytochemistry analysis in differentiation culturing conditions, OBs exhibited positive immunoreactivity for astrocytic cell marker anti-glial fibrillary acidic protein (anti-GFAP) and mature neuronal marker anti-microtubules associated protein 2 (anti-MAP2) Cellular nuclei were counterstained with Hoechst 33258 (blue); the bar value is 10 µm. (**D**) Graphical presentation of the invasion ability (number of invaded cells expressed in percentage) of Hu-OBNSCs2 to conditioned medoum (CM) derived from glioblastoma cancer cells (CM G-CSC) and to basal medium and FCS (DMEM + FCS) compared with those of WJ-MSC to the same above culturing media. * *p* < 0.05, ** *p* < 0.01 vs. “CM” groups. (**E**) Graphical presentation of the invasion ability of Hu-OBCSCs1 to CM G-CSC and to basal medium plus FCS (5%) compared with those of WJ-MSC to the same above culturing media. * *p* < 0.05, ** *p* < 0.01 vs. “CM” groups, *** *p* < 0.001.

**Figure 2 pharmaceutics-11-00045-f002:**
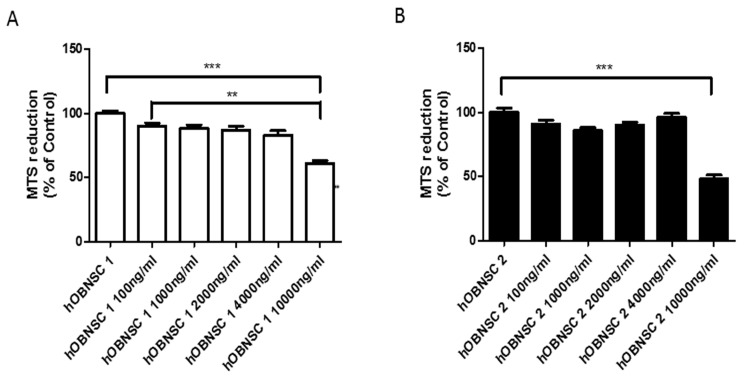
Sensitivity of Hu-OBNSCs to Paclitaxel (PTX). The histograms report the sensitivity of two Hu-OB cell lines (Hu-OBNSCs1 and Hu-OBNSCs2) to PTX evaluated at 24 h cytotoxicity assay with increasing concentration of PTX (from 100 to 10,000 ng/mL). The cell viability was taken as optical density (O.D) measured at 490 nm using a plate reader and expressed as percentage of control with (**) *p* value ≤ 0.01, and (***) *p* value ≤ 0.001. The mean ± SD of three experiments is reported.

**Figure 3 pharmaceutics-11-00045-f003:**
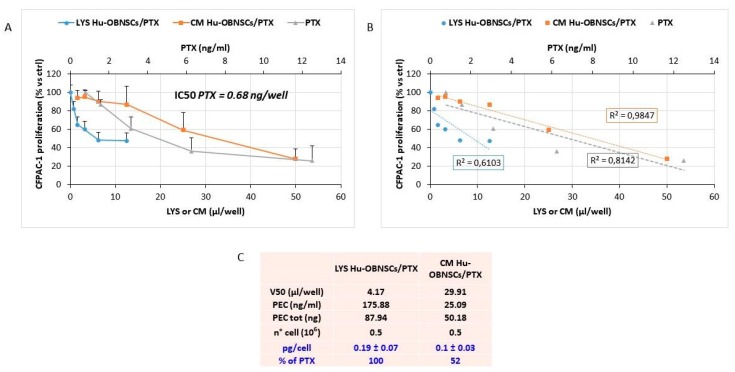
Incorporation and release of Paclitaxel by Hu-OBNSCs. (**A**) The effect of pure PTX, conditioned media (CM) and lysate (LYS) from Hu-OBNSCs treated with PTX (LYS Hu-OBNSCs/PTX, CM Hu-OBNSCs/PTX) is expressed as percentage of CFPAC-1 proliferation referred to that of untreated cells (100% proliferation). (**B**) The linear regression and the correlation coefficient (*r*^2^) of the dose response kinetics is reported. (**C**) The box reports the V50 value of LYS and CM (µL/well) used to estimate the amount of drug incorporated or released (PEC, ng/mL and PEC total, ng) by the Hu-OBNSCs/PTX cell. The percentage of drug incorporated and released by cell is expressed as pg/cell. Data are expressed as mean ± standard deviation (SD) of three independent experiments.

**Figure 4 pharmaceutics-11-00045-f004:**
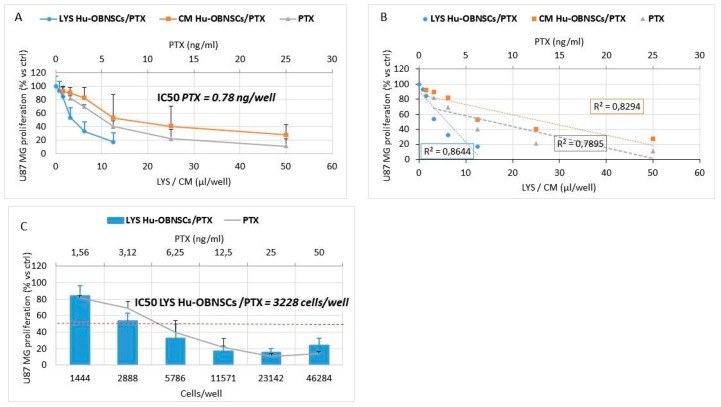
PTX release by primed Hu-OBNSCs is active against human glioma U87MG cells. (**A**) The effect of PTX, conditioned media (CM) and lysate (LYS) from Hu-OBNSCs treated with PTX (LYS Hu-OBNSCs/PTX, CM Hu-OBNSCs/PTX) is expressed as percentage of U87 MG proliferation referred to that of untreated cells (100% proliferation). (**B**) The linear regression and the correlation coefficient (*r*^2^) of the dose response kinetics is reported in figure. (**C**) The histogram reports the effect of LYS Hu-OBNSCs/PTX evaluated as number of cells/well in comparison to PTX (ng/mL) activity. The red dashed line represents the IC50 value in term of number of Hu-OBNSC/PTX cells/well. Values are expressed as mean ± standard deviation (SD) of three independent experiments.

**Figure 5 pharmaceutics-11-00045-f005:**
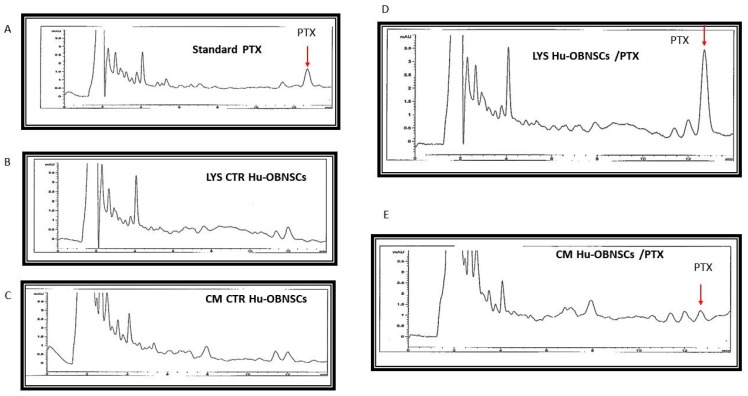
HPLC analysis of the Paclitaxel incorporated and released by Hu-OBNSCs. (**A**) HPLC chromatogram of standard drug (PTX = 1.000 ng/mL). The drug was eluted with a peak at 12.7 min (red arrow). (**B**,**C**) HPLC chromatogram of lysate and conditioned medium from untreated Hu-OBNSCs. (**D**,**E**) HPLC chromatograms of LYS and CM from Hu-OBNSCs treated with PTX showing a peak of identical retention time (red arrows). HPLC analysis revealed the presence of other nonspecific peaks (at 2.5–4 min) also present in the chromatogram of controls (**B**,**C**).

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
