# Peer review of "Human Olfactory Bulb Neural Stem Cells (Hu-OBNSCs) Can Be Loaded with Paclitaxel and Used to Inhibit Glioblastoma Cell Growth"

_pharmaceutics, 2019, doi:10.3390/pharmaceutics11010045_

Reviewer 1 Report

In this paper, the authors explored the paclitaxel uptake capacity of human olfactory bulb neural stem cells (hOBNSCs). The authors then presented convincing results showing that the primed OBNSCs are capable of inhibiting glioblastoma cell growth via release of paclitaxel. The authors envision this novel drug-delivery approach as a promising therapeutic tools for cancer therapy. I suggest the following revisions before publishing:

Introduction, line 44-46: Please cite literature to support this statement.

Materials and Methods, Sensitivity of Hu-OBNSCs1 and Hu-OBNSCs2 to Paclitaxel: Please explain the difference between OBNSCs1 and OBNSCs2. Were they isolated from separate donors?

Results, Sensitivity of h-OB cells to Paclitaxel: The cellular response to anticancer drug might not be restricted to metabolic activity. Cell apoptosis and cell cycle are also worth studying. 

Results, Efficiency of PTX incorporation and anticancer activity of PTX released by primed h-OB cells: So far the authors have only tested anticancer effect of the 24 hour conditioned medium. I suggest the authors also treat the cancer cells with CM collected from other time points (12 hour, 72 hour etc.), or run a indirect co-culture between the cancer cells and primed OBNSCs, for the extended drug release/anticancer effect of primed OBNSCs is a key selling point of this therapy compared to the lysate or PTX

It would be very interesting to see the actual homing and anticancer effect of primed OBNSCs in a mouse brain tumor model. This should be discussed as a future perspective of this work.

Author Response

Reviewer 1

Comments and Suggestions for Authors

In this paper, the authors explored the paclitaxel uptake capacity of human olfactory bulb neural stem cells (hOBNSCs). The authors then presented convincing results showing that the primed OBNSCs are capable of inhibiting glioblastoma cell growth via release of paclitaxel. The authors envision this novel drug-delivery approach as a promising therapeutic tools for cancer therapy. I suggest the following revisions before publishing:

Comment 1: Introduction, line 44-46: Please cite literature to support this statement.

Reply 1: We have added the following literature: [1] R. Mooney, M. Hammad, J. Batalla‐Covello, A. Abdul Majid, K.S. Aboody, Concise Review: Neural Stem Cell‐Mediated Targeted Cancer Therapies, Stem cells translational medicine, 7 (2018) 740-747.

Comment 2: Materials and Methods, Sensitivity of Hu-OBNSCs1 and Hu-OBNSCs2 to Paclitaxel: Please explain the difference between OBNSCs1 and OBNSCs2. Were they isolated from separate donors?

Reply 2:  The two Hu-OBNSCs, cultured  in vitro since 10 years, are derived from two separate donors. Recently,  in the lab we have conducted  a study on the Exome of those cell lines and others isolated in our lab. We expect to find significant specific and shared genetic variants between the two cell lines.

Comment 3:  Results, Sensitivity of h-OB cells to Paclitaxel: The cellular response to anticancer drug might not be restricted to metabolic activity. Cell apoptosis and cell cycle are also worth studying. 

Reply 3: The comment is right. Of course the mechanism of Paclitaxel on cancer cells has been extensively investigated and it is well known the block of cell cycle into G2/M.  The aim of this  part of our study was directed to confirm our previous reports cited that 2000 ng/ml of drug concentration do not affect significantly the  h-OB cells viability that is important to allow the cells to incorporate and release paclitaxel.

Comment 4:  Results, Efficiency of PTX incorporation and anticancer activity of PTX released by primed h-OB cells: So far the authors have only tested anticancer effect of the 24 hour conditioned medium. I suggest the authors also treat the cancer cells with CM collected from other time points (12 hour, 72 hour etc.), or run a indirect co-culture between the cancer cells and primed OBNSCs, for the extended drug release/anticancer effect of primed OBNSCs is a key selling point of this therapy compared to the lysate or PTX

Reply 4: Of course the reviewer’s suggestion is right. Many in vitro studies can be managed to further confirm the main  “proof of concept” that  OBNSCs are able to uptake and then release Paclitaxel (eg. to optimize time for incorporation and release, study the ratio primed cells/target cancer cells, etc…). We think that  to evaluate the PTX amount in the lysate and test its activity give us a preliminary simple but clear-cut  idea of the anti-cancer efficacy . By  considering the “in vivo “  future studies we think that the main important parameter is to quantify ( by HPLC on lysate) how much PTX is able to transport a single OBNSCs .From that will depend the decision on how many cells to use for treating the tumor. We must also consider that the drug primed cells will not be able to survive a  long time and all the internalized drug will be released in situ.

Comment 5:  It would be very interesting to see the actual homing and anticancer effect of primed OBNSCs in a mouse brain tumor model. This should be discussed as a future perspective of this work.

Reply 5:  The suggestion of the reviewer is very pertinent. The homing ability of normal or primed OBNSCs is a very important point  to progress from a “ proof of concept” into a possible clinical application. Of course this topic needs to be focused with “in vivo studies” that our laboratories hope to perform in a near future.

We added the following sentence in the discussion Of course this topic needs to be focused with “in vivo studies” that our laboratories hope to perform in a near future” lines: 553-554.

Reviewer 2 Report

Marei et al., present their findings in “Human olfactory bulb neural stem cells (Hu-OBNSCs) can be loaded with paclitaxel and used to inhibit glioblastoma cell growth” where they show that human olfactory bulb cells have the potential to be a carrier for paclitaxel and inhibit glioblastoma cell line or pancreatic cell line growth in vitro. While this is an interesting finding, a few questions remain, which if answered will significantly strengthen the findings outlined in this manuscript.

1.     Can the authors extend their findings to other glioblastoma cell lines and pancreatic cell lines to validate the effect observed in more than one system?

2.     How long do the Hu-OBNSCs survive in vivo? Can the authors evaluate the survival of transplanted Hu-OBNSCs in mice or in a 3D organoid system?

3.     Why is there such a stark discrepancy in the cytotoxic effect of pure Platinum (PTX) compared to lysate from Hu-OBNSCs loaded with PTX? Specifically why does the lysate from the Hu-OBNSCs have a higher potential to reduce CFPAC-1 proliferation compared to pure PTX? Can the authors comment on this?

4.      Furthermore the authors claim “The effect of the CM, Lysates and pure Paclitaxel against tumour cell proliferation was studied in 96 multiwell plates (Sarstedt, Germany) by using as targets U87GM and CFPAC-1.” – It would be required to study the effect of exposing the U87GM and CFPAC-1 to Hu-OBNSCs loaded with paclitaxel, not just the condition media and lysate.

5.     Can the authors discuss what advantage the loading the paclitaxel into Hu-OBNSCs would have over beads coated with the drug – especially in the case of transplant into a patient where introduction of cell lines is more contentious than implantable beads which will slowly release the drug?

Minor comments:

Figure 1 D, E to be redrawn and the fill of bars to be made more uniform as the legend is not easy to follow currently. Also, D and E to be interchanged in placements to ensure logical flow. Furthermore, the significance of the comparison in the left panel of 1E is currently confusing with 3 comparisons and only 2 sets of stars.

Author Response

Dear Referee,
Many thanks for your time and valuable comments! We have revised our manuscript according to the review reports step by step. Please check it.
Thanks again!
Best regards

Comments and Suggestions for Authors

Marei et al., present their findings in “Human olfactory bulb neural stem cells (Hu-OBNSCs) can be loaded with paclitaxel and used to inhibit glioblastoma cell growth” where they show that human olfactory bulb cells have the potential to be a carrier for paclitaxel and inhibit glioblastoma cell line or pancreatic cell line growth in vitro. While this is an interesting finding, a few questions remain, which if answered will significantly strengthen the findings outlined in this manuscript.

 1.     Can the authors extend their findings to other glioblastoma cell lines and pancreatic cell lines to validate the effect observed in more than one system?

A correctly suggested by the reviewer we could extend the investigation to many other cell lines! However our data  had the aim to demonstrate the main “proof of concept”  that Hu-OBNSCs are able to incorporate and  release PTX  and that the drug  maintains its anticancer activity. Although of interest, in our opinion, to demonstrate that PTX is active on other cell lines does not add news being this molecule yet extensively studied on many tumor models.

2.     How long do the Hu-OBNSCs survive in vivo? Can the authors evaluate the survival of transplanted Hu-OBNSCs in mice or in a 3D organoid system?

Our previous studies for the use of human OBNSC to treat Alzheimer’s, Parkinson’s, and spinal cord injuries have proved the ability of hOBNSC to survive at least for two months.

Therapeutic potential of human olfactory bulb neural stem cells for spinal cord injury in rats.

Marei HE, Althani A, Rezk S, Farag A, Lashen S, Afifi N, Abd-Elmaksoud A, Pallini R, Casalbore P, Cenciarelli C, Caceci T.

Spinal Cord. 2016 Oct;54(10):785-797. doi: 10.1038/sc.2016.14. Epub 2016 Feb 16.

Similar articles

Select item 2553654326.

Human olfactory bulb neural stem cells mitigate movement disorders in a rat model of Parkinson's disease.

Marei HE, Lashen S, Farag A, Althani A, Afifi N, A AE, Rezk S, Pallini R, Casalbore P, Cenciarelli C.

J Cell Physiol. 2015 Jul;230(7):1614-29. doi: 10.1002/jcp.24909.

Similar articles

Select item 2491117129.

Human olfactory bulb neural stem cells expressing hNGF restore cognitive deficit in Alzheimer's disease rat model.

Marei HE, Farag A, Althani A, Afifi N, Abd-Elmaksoud A, Lashen S, Rezk S, Pallini R, Casalbore P, Cenciarelli C.

J Cell Physiol. 2015 Jan;230(1):116-30. doi: 10.1002/jcp.24688.

   Of course  this  aspect  is essential to evaluate possible clinical applications. We hope to  enter a preclinical study in mice to answer the query and to evaluate the in vivo efficacy of the drug delivery by PTX primed OBNSCs. 

3.     Why is there such a stark discrepancy in the cytotoxic effect of pure Platinum (PTX) compared to lysate from Hu-OBNSCs loaded with PTX? Specifically why does the lysate from the Hu-OBNSCs have a higher potential to reduce CFPAC-1 proliferation compared to pure PTX? Can the authors comment on this?

I am sorry I do not understand this comment . The pure Paclitaxel (PTX)  was used as “reference drug”  in order to perform the biological dosage of PTX present in the lysate or in the conditioned medium according to an algorithm as described in M & M ( lines 185-190). The potential of lysate/CM to reduce CFPAC1 is directly proportional to the amount of PTX present.  Is without sense to speak about a “higher potential “ of lysate/CM in the respect of PTX.

4.      Furthermore the authors claim “The effect of the CM, Lysates and pure Paclitaxel against tumour cell proliferation was studied in 96 multiwell plates (Sarstedt,Germany) by using as targets U87GM and CFPAC-1.” – It would be required to study the effect of exposing the U87GM and CFPAC-1 to Hu-OBNSCs loaded with paclitaxel, not just the condition media and lysate.

To evaluate the PTX amount in the lysate and test its activity give us a preliminary simple but clear-cut  idea of the anti-cancer efficacy of this “ cell mediated delivery system”.  Also in prevision  of “in vivo “  future studies we think that the main preliminary important parameter will be quantify ( by HPLC on lysate) how much PTX is able to transport a single OBNSCs . This will help to select how many cells will be needed for treating a tumor mass. Of course , as correctly suggested by the  reviewer, to perform in vitro experiments of co-culture ( PTX primed  cells+cancer cells) could give further  functional information on the system. Unfortunately  in this study , aimed to sustain the concept we did not investigated the cell to cell interaction.

5.     Can the authors discuss what advantage the loading the paclitaxel into Hu-OBNSCs would have over beads coated with the drug – especially in the case of transplant into a patient where introduction of cell lines is more contentious than implantable beads which will slowly release the drug?

The reviewer’s comment  is  very pertinent. A lot of studies are directed to improve the drug bioavailability with many systems ( liposome, protein carrier, beads, etc).  Many of them did not have great success. The “cell mediated delivery” could be in effect  more difficult in term of regulatory. However we think that opens a new way at least for three aspects : a better integration of cells with the tumor environment, the possibility to educate in future the cell homing and the possible production of drug associated to EVs/exosomes. See lines 543-548

Minor comments:

 Figure 1 D, E to be redrawn and the fill of bars to be made more uniform as the legend is not easy to follow currently. Also, D and E to be interchanged in placements to ensure logical flow. Furthermore, the significance of the comparison in the left panel of 1E is currently confusing with 3 comparisons and only 2 sets of stars.

All requested corrections have been made, and new Fig 1 has been added.

Reviewer 3 Report

This manuscript dealt with the evaluation of Hu-OBNSCs to carry and release paclitaxel (PTX), producing effective cytotoxic effects against two types of cancer cells, human glioblastoma cell line and human pancreatic adenocarcinoma cell line. The article was well written and the data support the authors’ conclusion. Only several revisions need to be addressed.

#7 organization is missing.

Figure 1 A, B, C need higher resolution and the scale bars are missing. D & E need to be reversed.

Statistical significance is missing in Figure 2. The negative control group with no PTX is missing. The O.D. expression is not good as Percentage of Viable Cells if the authors have the correlation of O.D. and cell number.

Author Response

Dear Referee,
Many thanks for your time and valuable comments! We have revised our manuscript according to the review reports step by step. Please check it.
Thanks again!
Best regards

Comments and Suggestions for Authors

This manuscript dealt with the evaluation of Hu-OBNSCs to carry and release paclitaxel (PTX), producing effective cytotoxic effects against two types of cancer cells, human glioblastoma cell line and human pancreatic adenocarcinoma cell line. The article was well written and the data support the authors’ conclusion. Only several revisions need to be addressed.

 Comment 1: #7 organization is missing.

Reply 1: 7 IRCCS Istituto Ortopedico Galeazzi, Milan Italy   has been  added

Comment 2: Figure 1 A, B, C need higher resolution and the scale bars are missing. D & E need to be reversed.

 Reply 2: Corrected with new Figure 1

Comment 3: Statistical significance is missing in Figure 2. The negative control group with no PTX is missing. The O.D. expression is not good as Percentage of Viable Cells if the authors have the correlation of O.D. and cell number.

Reply 3: Done as suggested.

Round  2

Reviewer 2 Report

The authors have answered my previous comments satisfactorily

Reviewer 3 Report

Thank you very much for the revision.